# Outcome of adult cardiac surgery following COVID-19 infection in unvaccinated population in a national tertiary centre

Nur Aziah Ismail[ORCID]*, Ahmad Nazrin Jaapar, Alwi Mohamed Yunus, Abdul Rais Sanusi, Mohamed Ezani Taib, Mohd Azhari Yakub

Department of Cardiothoracic Surgery, Institut Jantung Negara, Kuala Lumpur, Malaysia

* dr.nurismail@ijn.com.my

## Abstract

### Background

Ever since COVID-19 was declared a pandemic, the world medical landscape has changed dramatically. As cardiac surgeons we not only have the duty to protect our patients and staff from COVID-19 infection, but we are also tasked with the responsibility to ensure those cardiovascular patients awaiting surgery are not harmed from an extended delay in surgery as the world comes to a halt from COVID-19. Currently there is limited literature on the outcome of cardiac surgery in the pre-operative Covid positive group. In this study we aim to assess the safety and outcome of patients undergoing cardiac surgery following Covid-19 infection.

### Patients and methods

This was a single centre retrospective observational study. All patients undergoing open heart surgery at Institut Jantung Negara from June 2020 to July 2021 were included in this study. Patients who were Covid positive pre-operatively were identified. Data from patient medical records collected contemporaneously were reviewed and analysed, supplemented by telephone call interviews after discharge.

### Results

2368 patients underwent open heart surgery from June 2020 until July 2021 in our centre. Of these, 0.5% (12 patients) were identified as Covid positive pre-operatively. Mean age of patients were 59.1 ± 14.8 years old. Mean Ejection Fraction was 46.4 ± 12.9. Most patients (75%) were asymptomatic with covid infection and only one patient were admitted to hospital for Covid infection. Mean duration from Covid PCR positive swab to surgery were 46.3 ± 32.7days. Most of the patients (66.7%) underwent operation on an emergency or urgent basis. Median time to extubation was 1 day. Median ICU length of stay was 1 day. 25% patients required non-invasive ventilation post-operatively and one patient was discharged home on long term oxygen therapy. There were 2 deaths- none of which were covid related mortality.

**Data Availability Statement:** The minimal data set is available here: 10.6084/m9.figshare.19538236.

**Funding:** The author(s) received no specific funding for this work.

**Competing interests:** The authors have declared that no competing interests exist.

## Conclusion

Cardiac surgery could be performed safely in patients with pre-operative Covid-19 infection after a period of recovery, especially in the asymptomatic to mild category of infection. Multi-disciplinary team approach may be useful in deciding the timing of surgery for complex cases.

## Introduction

On March 11[th] 2020, COVID-19 was declared a pandemic by the World Health Organization. Since then, the world medical landscape has changed dramatically. Elective surgeries were put on hold to pour resources such as intensive care units, ventilators, and healthcare workers for COVID-19 patients. Countries that were previously unaffected are currently experiencing mounting rates of the COVID-19 infection with associated increases in COVID-19-related deaths. Now almost two years later, despite vaccination, we are seeing an exponential increase of the infection, mainly due to the more aggressive Delta variant.

As cardiac surgeons we not only have the duty to protect our patients and staff from COVID-19 infection, but we are also tasked with the responsibility to ensure that those cardio-vascular patients awaiting surgery are not harmed from an extended delay in surgery as the world comes to a halt from COVID-19. It has been reported that for patients awaiting coro-nary surgery, median waiting list mortality rates were 2.6% per month, with mortality risk increasing 11% per month, and 12% patients experiencing myocardial infarction while on the waiting list [1, 2]. Meanwhile, reported mortality rate while waiting for aortic valve replace-ment for aortic stenosis can be as high as 3.7% at one month and 11.6% at six months [3].

As the number of COVID-19 patients continue to rise, we would expect to be operating on more patients with previous or recent COVID-19 infection. However, there is limited litera-ture on this cohort of patients on the cardiac surgery side [4, 5]. It is unknown whether patients recently recovered from COVID-19 are more susceptible to post-operative complica-tions, as their angiotensin-converting-enzyme 2, which may be protective against acute lung injury, may be consumed during COVID-19 infection [6–9]. The largest study to date on the topic was by Sanders and colleagues in which they reported the outcome of cardiac surgery in 17 patients with pre-operative COVID-19 infection- although there was no mention of dura-tion from diagnosis of COVID-19 infection to surgery [10]. In this study we aim to assess the safety and outcome of patients undergoing cardiac surgery following recent Covid-19 infection.

## Methods

All patients undergoing elective, urgent and emergency open heart surgery in National Heart Institute (IJN) Kuala Lumpur, Malaysia from 1[st] July 2020 to 1[st] July 2021 were included in this study. Patients were identified from the hospital cardiac surgery database. All pre-opera-tive patients required routine pre-operative COVID-19 screening via Polymerase Chain Reac-tion (PCR) swab test either at our local hospital or at the referring/district hospital. Patients who were diagnosed with COVID-19 infection via positive swab test pre-operatively, were identified from our institute's COVID database and their cases were reviewed. This time period was prior to vaccination of the mass population and hence our cohort of patients at this time were all non-vaccinated patients. Data from electronic patient records and patients

medical records collected contemporaneously were reviewed retrospectively. Phone call interviews were carried out to supplement data. This study received the approval of IJN Research Committee (IJNREC) (Project registration ID: IJNREC/523/2021). Consents were obtained verbally from all human subjects in thus study.

## Results

During the study period, we performed 2368 open heart surgery. Twelve of these patients (0.5%) were diagnosed with COVID-19 infection pre-operatively. Mean age of patient was 59.1 ± 14.8 years with equal number of male and female patients. 91.7% of the patients were in NYHA Class I- II pre-operatively, and only one patient was in NYHA Class III-IV. Half of the patients were in CCS 0 angina status, one patient was in CCS 1–2 and the remaining five patients (41.7%) were in CCS 3–4. Mean Euroscore II were 4.59 ± 3.97. More than half of these patients had hypertension (58.3%, n = 7), half of the patients had type 2 Diabetes mellitus, a third of these patients had Chronic Kidney Disease (CKD) and one patient was in endstage renal failure (ESRF) requiring dialysis. With regards to pre-operative cardiac diagnosis, five patients (41.7%) had isolated triple vessel coronary artery disease with or without left main stem disease, two patients had valvular heart disease, three patients had combined coronary artery disease and valvular heart disease, one patient had infective endocarditis and one patient had ascending aortic aneurysm with coronary artery disease. Mean pre-operative ejection fraction was 46.4% ± 12.9% (Table 1).

With regards to pre-operative Covid status, 75% (n = 9) were asymptomatic with COVID-19 infection, two patients (16.7%) had fever and flu like symptoms, and only one patient (8.3%) had dyspnoea requiring hospital admission and oxygen requirement. None of the patients required ICU admission or intubation for COVID-19 infection. A third of these patients were infected in the community, two patients contracted COVID-19 infection in the local/district hospital during initial admission with cardiac complaints. The remaining half of the patients' source of COVID-19 infection was undetermined/ untraceable. Following the

**Table 1. Patients demographics.**

| Demographics | | Total N (%) |
|---|---|---|
| **Gender** | Male | 6 (50%) |
| | Female | 6 (50%) |
| **Age** | Mean +/- SD | 59.1 +/- 14.8 |
| **NYHA Status** | 1–2 | 11 (91.7) |
| | 3–4 | 1 (8.3) |
| **Angina Status** | CCS 0 | 6 (50.0) |
| | CCS 1–2 | 1 (8.3) |
| | CCS 3–4 | 5 (41.7) |
| **Co-morbidities** | Hypertension | 7 (58.3) |
| | Diabetes Mellitus | 6 (50.0) |
| | Chronic Kidney Disease (CKD) | 4 (33.3) |
| | End Stage Renal failure (ESRF) | 1 (8.3) |
| **Cardiac Diagnosis** | Triple Vessel Disease +/- Left Main Stem | 5 (41.7) |
| | Valvular Heart Disease | 2 (16.7) |
| | Coronary artery disease + Valvular heart disease | 3 (25.0) |
| | Infective endocarditis | 1 (8.3) |
| | Ascending aortic aneurysm | 1 (8.3) |
| **Ejection Fraction** | Mean +/- SD | 46.4 +/- 12.9 |

**Table 2. Operation and urgency.**

| | | Total N (%) |
|---|---|---|
| **Duration from Covid diagnosis to Operation (days)** | Mean +/- SD | 46.3 ± 32.7 |
| | Median (Q1, Q3) | 36.5 (27.0, 56.3) |
| **Operation Urgency** | Urgent & Emergency | 8 (66.7%) |
| | Elective | 4 (33.3) |
| **Operation** | Isolated CABG | 6 (50) |
| | CABG + valve replacement | 2 (16.7) |
| | Valve replacement | 2(16.7) |
| | Aortic surgery | 1 (8.3) |
| | Redo Valve replacement | 1 (8.3) |
| **Cardiopulmonary bypass (CPB) time (mins)** | Median (Q1, Q3) | 122 (79, 169) |
| **Cross-Clamp time (mins)** | Median (Q1, Q3) | 93 (72, 136) |

Covid diagnosis via the positive COVID-19 PCR swab test, the mean waiting time to surgery amongst these patients was 46.3 ± 32.7 days (14–136). Pre-operative lung function test were normal in majority of the patients 75% (n = 8) and the remaining 25% (n = 4) had restrictive lung defect pattern on spirometry. Three patients had consolidation on Chest X-Ray pre-operatively, whilst the rest had normal chest Xray.

Majority of these patients had surgery on an urgent or emergency basis (66.7%, n = 8). The main operation undertaken were isolated Coronary Artery Bypass Grafting (CABG) n = 6. Two patients had combined procedure of CABG + valve replacement, another two patients had isolated valve replacements, one patient had replacement of ascending aorta, and one patient had redo valve surgery. Median cardiopulmonary bypass (CPB) time was 122mins, and median cross clamp time was 93 minutes (Table 2).

Post-operatively, five patients were extubated within 24 hours of surgery, three patients were extubated on day 1 post-operativelyand two patientswere extubated on post-operative day 2. None of the patients required re-intubation; however, 25% (n = 3) required non-invasive ventilation in the form of CPAP/BiPAP after extubation. Respiratory failure occured in 25% of the patients (n = 3). Out of these three patients, one required long term oxygen therapy upon discharge from hospital. There were two patients that developed acute kidney injury (AKI) on pre-existing CKD. Both patients made recovery to baseline renal function upon discharge from the hospital. One patient developed heart failure post-operatively requiring prolonged inotropic support, however, this patient had poor left ventricular function pre-operatively (Ejection fraction <30%).

Seven patients had less than 2 days stay in the intensive care unit (ICU), four patients spent 3–7 days in ICU, and one patient spent more than a week in the ICU. Median length of stay for ICU was 2 days. Median length of stay in the hospital post-operatively, prior to discharge home, was 9 days. 25% (n = 3) of the patients spent less than a week post-operatively in hospital, 50% of the patients (n = 6) spent 8–14 days in the hospital post-operatively, and only one patient required hospitalization beyond 2 weeks. Causes of prolonged length of stay in our patients were due to respiratory wean, treatment of heart failure, warfarinisation and achieving target INR, and treatment of AKI (Table 3).

There were two mortality (16.67%), whilst the rest of the patients were discharged home well. None of the mortality were COVID-19 related deaths. The first mortality was a 45-year-old male patient with a Euroscore II of 5.8. He had a background of triple vessel disease and severe mitral regurgitation and ESRF on haemodialysis. He underwent urgent CABG and mitral valve replacement 26 days after he tested positive on COVID-19 PCR swab. Pre-

**Table 3. Post-operative outcome.**

| | | Total N (%) |
|---|---|---|
| **Time to extubation** | <24 hours | 5 (41.6) |
| | 1day | 3 (25.0) |
| | ≥2 days | 2 (16.7) |
| **Post-operative complication** | Non Invasive Ventilation (NIV) | 3 (25) |
| | Respiratory failure | 3 (25) |
| | Re-intubation | 0 |
| | Acute Kidney Injury | 2 (16.7) |
| | Heart Failure | 1 (8.3) |
| **Length of Stay in Hospital** | ≤ 7 days | 3 (25) |
| | 8–14 days | 6 (50) |
| | >14 days | 1 (8.3) |
| **Outcome** | Alive | 10 (83.3) |
| | Dead | 2 (16.7) |

operatively he was asymptomatic with regards to his covid infection, although his pre-operative Chest X-Ray showed consolidation in the right mid and lower zone. He developed sepsis post-operatively and pseudomonas was isolated in his blood culture. He died of sepsis and multi-organ failure on day six after the operation. The second mortality was a 61 year old female patient with a previous double valve replacement in 2007. She returned for redo aortic valve replacement 60 days after she tested positive for COVID-19 infection. She was asymptomatic for her COVID infection and her pre-operative Chest X-Ray was normal. She died due to myocardial failure post re-do aortic valve replacement on post-operative day one.

## Discussion

COVID-19 pandemic has caused significant mortality and morbidity to the general population [11]. Patients undergoing cardiac surgery are a unique population in this COVID-19 era due to the risk of exposure to others from highly invasive, aerosol generating procedure, the potentially prolonged hospitalization or ICU stay and the overall intense healthcare resource use [12]. The potential worsening of the infection due to systemic inflammatory response from the use of cardiopulmonary bypass machine renders it essential to identify those who are COVID-19 positive pre-operatively. Since our centre is a national standalone tertiary cardiac hub that continued to run elective cardiac surgery services despite the pandemic, a protocol was implemented early in the pandemic to screen all patients undergoing cardiac surgery for COVID-19 infection. COVID-19 screening was done either at the time of referral from district hospital and again within 48hours pre-operatively. This is to ensure that we do not operate on patients with recent COVID-19 infection and to protect our healthcare workers and other patients from the infection.

ASA-APSF Joint statement recommended for elective surgery patients to wait at least 4 weeks before operating on asymptomatic or patients with mild non respiratory symptoms and at least 6 weeks wait for symptomatic patients who did not require hospitalization, at least 8–10 weeks for symptomatic patient who were hospitalized [13]. Most of our patients were asymptomatic with COVID-19 infection pre-operatively and only one patient was hospitalized for COVID-19 infection. The mean duration from COVID PCR swab to surgery for our patients was 46.3 days i.e., average of a little over six weeks. This could explain the decent post-operative outcome in our cohort of patients. This finding confirmed the findings of Sanders

et al. who conducted a retrospective review of 9 UK cardiac surgery centres. They studied 17 patients with pre-operative COVID-19 diagnosis and concluded that these cohort of patients with pre-operative COVID-19 diagnosis recovered in a similar way to non-COVID-19 patients [10]. In keeping with their findings, we had no COVID-19 related mortality post-operatively. Similarly, our mean time to extubation period of 1 day. However our hospital length of stay post-operatively were 2 days longer compared to theirs (9 vs 7 days), this could be explained by our patients having higher Euroscore II 4.6% vs 2.8% in their cohort.

The patients in our cohort who developed post-operative pulmonary complication requiring non-invasive ventilation were operated within a median of 24 days from COVID-19 diagnosis. The reason for earlier surgery in these cases were due to the more urgent nature of their underlying cardiac disease. The main issue with operating within the four weeks within the diagnosis of COVID-19 infection is the risk of mortality and pulmonary complication rates. Recent evidence suggests that COVID-19 patients who undergo surgery may be more susceptible to pneumonia and ARDS post-operatively, even when they are asymptomatic [14]. Patients who had surgery less than six weeks after COVID-19 diagnosis had significantly higher adjusted 30-day postoperative pulmonary complication rates compared to patients who did not have COVID-19 infection. This risk returns to baseline after seven weeks [15]. COVID-Surg Collaborative based on its large multinational study of greater than 100,000 patients across all surgical specialty, suggested that elective surgery should not be scheduled within 7 weeks of diagnosis of Covid-19 infection, unless the risk of deferring surgery outweigh the risk of post-operative morbidity/mortality associated with COVID-19 [16].

Interestingly we noted that a quarter of our patients had restrictive lung defect on spirometry although these group did not develop post-operative pulmonary complications. This suggests that spirometry may not be the best tool to predict post-operative pulmonary complication. We also noted that despite being asymptomatic, some patients had abnormal Chest X-ray pre-operatively. The patients that developed post-operative pulmonary complications had seemingly normal chest X-Ray pre-operatively, but a clear picture of organizing pneumonia in the post-operative CT scan. This sparked the question of whether this cohort of patients would benefit from pre-operative CT scan of the thorax to risk stratify and guide timing of surgery. Further research should be directed into this area.

The limitations we faced in this study were that most of our patients were asymptomatic and had mild COVID-19 infection, therefore this study did not include the group of patients that are more severely affected with COVID-19 infection. This was also a retrospective study. Furthermore, this study was conducted during the period of time when vaccination was still novel, and the mass population was far from herd immunity. The outcome could be different in vaccinated group of population who were infected with COVID-19 pre-operatively. This would be another area to explore for future direction of research.

At the time this study was conducted, most of the available guidelines were based on patients in the general surgical group, which were somewhat different to cardiac surgery group, and the guidelines were mainly for elective surgery rather than urgent surgery. Cardiac surgery patients require a delicate balance between waiting to minimize peri-operative mortality and morbidity risk versus waiting too long that they succumbed to their cardiovascular disease. A multi-disciplinary team (MDT) approach with cardiologist, respiratory physician, intensivist and infectious disease team could prove useful in this scenario. On this note, our institution has recently implemented a protocol for our pre-operative COVID-19 patients-this includes a guide on the timing of surgery, the implementation of the MDT approach and pre-operative workup in an effort to minimise risk of post-operative mortality and morbidity (Fig 1).

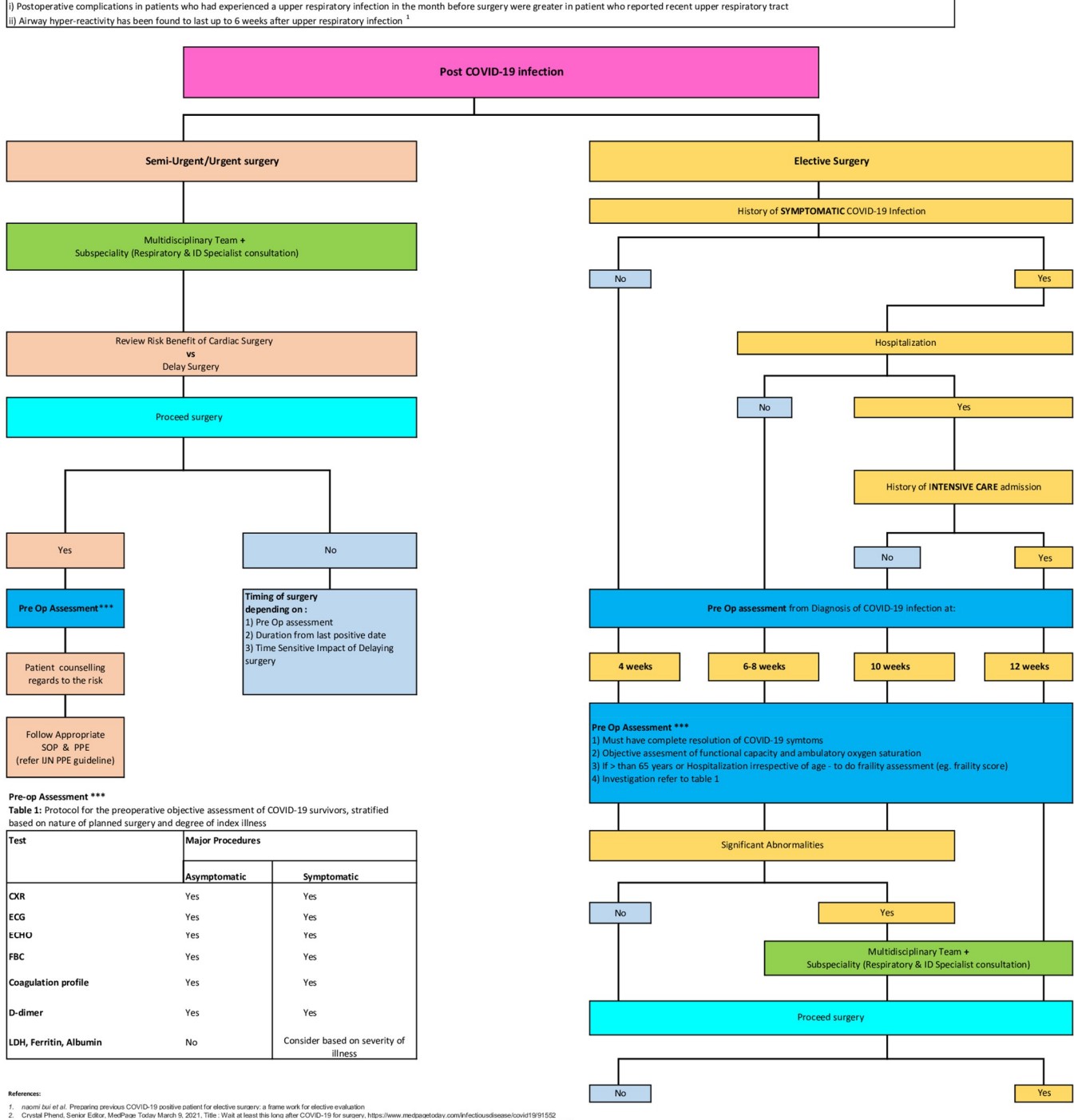

**Fig 1. Framework for pre-operative evaluation for patients with pre-op COVID-19.**

COVID-19 pandemic continues to be an evolving arena, with which we are constantly learning, researching and new knowledge brought to light, and guidelines changed as a result. This study confirms the findings of current literature. Further research into the role of CT scans in these cohorts of patients to guide timing of operation and minimise post-operative pulmonary complications could prove to be useful. The role of vaccination in these group of patients should also be explored.

## Conclusion

Cardiac surgery could be performed safely in patients with pre-operative Covid-19 infection after a period of recovery, especially in the asymptomatic to mild category of infection. Multi-disciplinary team approach may be useful in deciding the timing of surgery for complex cases.

## Author Contributions

**Conceptualization:** Alwi Mohamed Yunus, Mohd Azhari Yakub.

**Data curation:** Nur Aziah Ismail, Ahmad Nazrin Jaapar.

**Formal analysis:** Nur Aziah Ismail, Abdul Rais Sanusi.

**Investigation:** Nur Aziah Ismail, Ahmad Nazrin Jaapar, Abdul Rais Sanusi.

**Methodology:** Nur Aziah Ismail, Abdul Rais Sanusi.

**Resources:** Mohamed Ezani Taib, Mohd Azhari Yakub.

**Supervision:** Alwi Mohamed Yunus, Abdul Rais Sanusi, Mohd Azhari Yakub.

**Validation:** Abdul Rais Sanusi, Mohd Azhari Yakub.

**Writing – original draft:** Nur Aziah Ismail.

**Writing – review & editing:** Nur Aziah Ismail, Alwi Mohamed Yunus, Mohamed Ezani Taib, Mohd Azhari Yakub.

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
