## [Decision Letter · Decision Letter 0]

5 Jan 2022

PONE-D-21-33226Outcome of cardiac surgery following recent COVID-19 infection in unvaccinated patientsPLOS ONE

Dear Dr. Ismail

Thank you for submitting your manuscript to PLOS ONE. After careful consideration, we feel that it has merit but does not fully meet PLOS ONE’s publication criteria as it currently stands. Therefore, we invite you to submit a revised version of the manuscript that addresses the points raised during the review process.

We look forward to receiving your revised manuscript.

Kind regards,

Alessandro Leone, MD

Academic Editor

PLOS ONE

Journal Requirements:

Reviewers' comments:

Reviewer's Responses to Questions

**Comments to the Author**

1. Is the manuscript technically sound, and do the data support the conclusions?

Reviewer #1: Yes

Reviewer #2: Partly

Reviewer #3: Yes

2. Has the statistical analysis been performed appropriately and rigorously? 

Reviewer #1: Yes

Reviewer #2: Yes

Reviewer #3: I Don't Know

3. Have the authors made all data underlying the findings in their manuscript fully available?

Reviewer #1: Yes

Reviewer #2: Yes

Reviewer #3: Yes

4. Is the manuscript presented in an intelligible fashion and written in standard English?

Reviewer #1: Yes

Reviewer #2: Yes

Reviewer #3: Yes

5. Review Comments to the Author

Reviewer #1: Dear the authors of the manuscript entitled "Outcome of cardiac surgery following recent COVID-19 infection in unvaccinated patients"

I was glad reading this manuscript which reported the outcomes of cardiac surgery in patients with previous infection of COVID-19.

I do agree with the authors about the importance of this study despite the limited numbers of patients included in this study, however it could be an addition to the literature discussing this issue

I have couple of points here to mention:

1. The period of in-hospital stay was longer than the counterpart patients, is there any explanation?..Is there any reporting of the period and amount of chest tube drainage? Because we noticed a prolonged periods of chest tube drainage in such patients after cardiac surgery

2. Do you utilize chest ct Scan before cardiac surgery to make sure that these patients are ready to undergo surgery?

3. Was there any special management protocols for patients during the cardiac surgery procedure, that is different from the other patients,? ie anticoagulation protocol, steroid,and antibiotic regimenens?

Thank you

Reviewer #2: This observational and descriptive report was done in a busy cardiac center with a large sample size, which was a highly selected cohort from referral and preop selection. It might be too premature to draw the interim conclusion that cardiac surgery in such patients because there were obvious selection biases. There are several issues here.

1. Even with preop COVID testing, how could there were 0.5% still receiving cardiac surgery in which there some elective cases? Please explain. Were they missed so still undergoing non-emergent cardiac surgery? In a well-managed hospital, unless life-threatening, there should 0% non-emergent cardiac surgery in this pandemic. Please justify and explain.

2. This study was done in the national specialized heart center which has the best faculty, staff, and facilities to achieve the non-inferior outcomes in a highly selected cohort. There are lots of lower-level cardiac surgery centers and they may not achieve this good outcome. If the denied COVID cases had received the cardiac surgery, the outcomes might have not been so favorable. If the data were collected in multi-centers, regional data, national data, or even international data registries, the conclusion would be much more convincing. The authors please explain or justify their highly selected patients could represent the real world.

3. Most of the presented were adult coronary cases, which were just a part of "cardiac surgery" which includes valve, aortic, root, transplant, congenital, etc. There were no pediatric or transplant cases. The data shown were mostly coronary cases and hence were not representative for "cardiac surgery". Please revise the Title to better reflect the case characteristics.

Therefore, the authors need to justify why there were still non-emergent COVID cases undergoing cardiac surgery, explain the case being representative to the outside world, revise the title to reflect the case majority of adult and coronary cases (no pediatric, no transplant), etc.

Reviewer #3: During such a difficult times of pandemic, in time of uncertanty regarding the influence of the still novel COVID-19, such kind of research makes it easier to health care professionals to conduct their daily activities and provide best possible care. This paper outlines the most important factors regarding treating COVID-19 patients having a cardiovascular ongoing disease. The importance of the research also lies under the fact that not much work has been done in this direction, there is not enough data, so the level of frustration is high when treating patient with COVID-19 previous infection. This paper will help health care professionals treating such kind of patients on a daily basis, the information provided can help predict outcomes after cardiac surgery in this group of patients. A lot of opinions has been heard, but not many are evidence based and statistically strong enough to take into consideration. It is clear from this work that cardiac surgical procedures could be safely done in pre-operative Covid-19 infection group after a period of recovery, and success rate is higher in the asymptomatic or mild category of infection.

This study assessed well the safety and outcomes of the cardiac surgery in the pre-operative Covid positive group. All the patients went through a strict selection criteria, every patient was Covid screened before surgeries. Out of 2368 patients 12 patients were identified as a pre-operative covid positive patients who went through a period of recovery. This paper shows that cardiac surgical procedure outcomes are better in a group of patients where more than 4 weeks has passed after the infection, and also the results are better in asymptomatic or mild category. The timeframe here is very important, because this will help the team the decide and choose the most appropriate timing for the surgery, of course taking into consideration the severity of the condition. Covid infection can halt the surgery timing, thus increasing the mortality, this was addressed in this paper, which is important. In my opinion 12 patients is enough to get some sentence of what is going on with covid patients undergoing cardiac surgical procedure, but seeing more patients in the study would strengthen the results.

The paper merits publication , is well-written and well organized.

6. PLOS authors have the option to publish the peer review history of their article (what does this mean?). If published, this will include your full peer review and any attached files.

Reviewer #1: **Yes: **Salah Eldien Altarabsheh

Reviewer #2: **Yes: **Robert J. Chen, MD, MPH

Reviewer #3: No

---

## [Author Response · Author response to Decision Letter 0]

17 Feb 2022

Dear reviewers, 

Many thanks for taking the time to read this paper and for your comments. Please find our response to this comment below: 

Response to Reviewer #1: 

Many thanks for your time and feedback. 

1. Question: The period of in-hospital stay was longer than the counterpart patients, is there any explanation?..Is there any reporting of the period and amount of chest tube drainage? Because we noticed a prolonged periods of chest tube drainage in such patients after cardiac surgery

Answer: With regards to the longer in-hospital stay, this is due to respiratory wean, treatment of heart failure, warfarinisation and achieving target INR, and treatment of AKI. Please refer to table 3 and Paragraph #2 under the Discussion. With regards to comparison of length of hospital stay with the western world, we are comparing two separate centres from two different parts of the world with a different cohort of patient demographics, referral pathway, access to healthcare and resources. This may also explain why our cohort has a higher Euroscore II compared to their population. 

Question: Is there any reporting of the period and amount of chest tube drainage? Because we noticed a prolonged periods of chest tube drainage in such patients after cardiac surgery

Answer: This is an interesting observation. We did not specifically look at drain output in this study, however from observation we did not notice any change in terms of drain output compared to patients without prior covid infection. 

2. Question: Do you utilize chest ct Scan before cardiac surgery to make sure that these patients are ready to undergo surgery?

Answer: This is a good point raised. This study was conducted during early in the pandemic whereby there were no clear guideline on how to go about operating on these cohort of patients or knowledge of what was necessary for the work up of these patients. At that point we were only reliant on chest-Xray and lung function testig for these patients. Later on during the pandemic as we have had more experience in dealing with pre-op covid-19, we noticed that a normal Chest Xray doesn’t preclude the need for chest CT scan. This is further reinforced by the findings of this study. This has been addressed in the discussion part of this paper, and we have since then changed our practice at the end of last year, to include pre-op CT chest for patients with prior COVID-19 infection. I agree, it would be interesting to see the findings of pre-op CT and see whether it correlates with post-operative respiratory morbidity for these patients. 

3. Question: Was there any special management protocols for patients during the cardiac surgery procedure, that is different from the other patients,? ie anticoagulation protocol, steroid, and antibiotic regimens?

Answer: This was very much dependent on the patient status. i.e. if the patient showed pre-operative restrictive lung defect on spirometry, we would go more gentle on weaning the ventilation vs early extubation for normal patients. In terms of anticoagulation, steroids and antibiotics, we did not do anything differently for these patients. 

Many thanks. 

Reviewer 2 comments : 

Reviewer #2: This observational and descriptive report was done in a busy cardiac center with a large sample size, which was a highly selected cohort from referral and preop selection. It might be too premature to draw the interim conclusion that cardiac surgery in such patients because there were obvious selection biases. There are several issues here.

1. Even with preop COVID testing, how could there were 0.5% still receiving cardiac surgery in which there some elective cases? Please explain. Were they missed so still undergoing non-emergent cardiac surgery? In a well-managed hospital, unless life-threatening, there should 0% non-emergent cardiac surgery in this pandemic. Please justify and explain.

2. This study was done in the national specialized heart center which has the best faculty, staff, and facilities to achieve the non-inferior outcomes in a highly selected cohort. There are lots of lower-level cardiac surgery centers and they may not achieve this good outcome. If the denied COVID cases had received the cardiac surgery, the outcomes might have not been so favorable. If the data were collected in multi-centers, regional data, national data, or even international data registries, the conclusion would be much more convincing. The authors please explain or justify their highly selected patients could represent the real world.

3. Most of the presented were adult coronary cases, which were just a part of "cardiac surgery" which includes valve, aortic, root, transplant, congenital, etc. There were no pediatric or transplant cases. The data shown were mostly coronary cases and hence were not representative for "cardiac surgery". Please revise the Title to better reflect the case characteristics.

1. Question: Even with preop COVID testing, how could there were 0.5% still receiving cardiac surgery in which there some elective cases? Please explain. Were they missed so still undergoing non-emergent cardiac surgery? In a well-managed hospital, unless life-threatening, there should 0% non-emergent cardiac surgery in this pandemic. Please justify and explain.

Answer: These 0.5% of patients who had cardiac surgery has had Covid in the past AND recovered from Covid-19 prior to undergoing surgery, they were not Active COVID cases. Please refer to Table 2 in the result sectionwhich states mean duration from Covid to Operation for these patients were 46.3 days, i.e. 6 weeks. 

We are the national referral centre at this time, whereby all other units in the country have closed their doors to the cardiac surgery population in order to provide COVID -services, we have been designated as the national cardiac hub and a “clean” centre and continued to provide services for the cardiac population of this country. Therefore we received, all emergency and urgent cases in this time as we were the only cardiac hub still operating at this time. Out of these 12 patients, two- thirds of these were urgent and emergency cases, and another one-third were patients who’s had their elective operations delayed for a while due to the pandemic and started having increasing symptoms. As you are aware, COVID-19 came in waves, and has been around for two years now. During the period of surges of cases and lockdowns we limit the operation to urgent and emergency only, and during period of recovery we re-open our elective services as per the institutional and national policy. As a national cardiac centre, we could not afford to completely close our elective services such as CABGs and AVRs for 2 years, as these would have grave consequence on these cohort of patients. Many of the elective cases during these times already had their operation delayed up to one year. The decision on which elective cases to be operated on, were purely clinical according to their disease progress. i.e. increasing frequency in angina, increasing shortness of breath, etc. 

2. Question: This study was done in the national specialized heart center which has the best faculty, staff, and facilities to achieve the non-inferior outcomes in a highly selected cohort. There are lots of lower-level cardiac surgery centers and they may not achieve this good outcome. If the denied COVID cases had received the cardiac surgery, the outcomes might have not been so favorable. If the data were collected in multi-centers, regional data, national data, or even international data registries, the conclusion would be much more convincing. The authors please explain or justify their highly selected patients could represent the real world.

Answer : All the other cardiac centers at this time period were not carrying out cardiac surgery as they had to accommodate COVID-19 in their ICU and therefore deemed unsafe to carry out cardiac surgery as per the national policy at the time. Therefore all these patients were diverted to our centre. Hence we are not able to provide you with data from other centres during that point of time. With regards to your point of whether this is representative of real world data, please refer to the discussion part where we have compared our findings with authors in the UK which have done this study in a multi-centre, and our result is much in keeping with theirs. 

3. Question: Most of the presented were adult coronary cases, which were just a part of "cardiac surgery" which includes valve, aortic, root, transplant, congenital, etc. There were no pediatric or transplant cases. The data shown were mostly coronary cases and hence were not representative for "cardiac surgery". Please revise the Title to better reflect the case characteristics.

Answer: During this period of time, we had no cardiac or pulmonary transplant done in this centre or any other centre in the country. This was the Institution and national policy during this pandemic, as all resources were diverted to support the pandemic. This study only looks at adult cardiac surgery practice- hence we have changed the title to define the population we represent in this paper. 

Reviewer #3 comments: 

During such a difficult times of pandemic, in time of uncertainty regarding the influence of the still novel COVID-19, such kind of research makes it easier to health care professionals to conduct their daily activities and provide best possible care. This paper outlines the most important factors regarding treating COVID-19 patients having a cardiovascular ongoing disease. The importance of the research also lies under the fact that not much work has been done in this direction, there is not enough data, so the level of frustration is high when treating patient with COVID-19 previous infection. This paper will help health care professionals treating such kind of patients on a daily basis, the information provided can help predict outcomes after cardiac surgery in this group of patients. A lot of opinions has been heard, but not many are evidence based and statistically strong enough to take into consideration. It is clear from this work that cardiac surgical procedures could be safely done in pre-operative Covid-19 infection group after a period of recovery, and success rate is higher in the asymptomatic or mild category of infection.

This study assessed well the safety and outcomes of the cardiac surgery in the pre-operative Covid positive group. All the patients went through a strict selection criteria, every patient was Covid screened before surgeries. Out of 2368 patients 12 patients were identified as a pre-operative covid positive patients who went through a period of recovery. This paper shows that cardiac surgical procedure outcomes are better in a group of patients where more than 4 weeks has passed after the infection, and also the results are better in asymptomatic or mild category. The timeframe here is very important, because this will help the team the decide and choose the most appropriate timing for the surgery, of course taking into consideration the severity of the condition. Covid infection can halt the surgery timing, thus increasing the mortality, this was addressed in this paper, which is important. In my opinion 12 patients is enough to get some sentence of what is going on with covid patients undergoing cardiac surgical procedure, but seeing more patients in the study would strengthen the results.

The paper merits publication , is well-written and well organized.

Answer: Many thanks for taking the time to read and for your positive comments and feedback. We hope this data will contribute to future decision-making with regards to timing and to further inform patients during consent taking process for these cohort of patients. Thanks again.

---

## [Editor Report · Decision Letter 1]

14 Mar 2022

Outcome of adult cardiac surgery following recent COVID-19 infection in unvaccinated patient population in a national tertiary centre

PONE-D-21-33226R1

Dear Dr.Nur Aziah Ismail

We’re pleased to inform you that your manuscript has been judged scientifically suitable for publication and will be formally accepted for publication once it meets all outstanding technical requirements.

Kind regards,

Alessandro Leone, MD

Academic Editor

PLOS ONE
---

## [Editor Report · Acceptance letter]

31 Mar 2022

PONE-D-21-33226R1 

Outcome of adult cardiac surgery following COVID-19 infection in unvaccinated population in a national tertiary centre. 

Dear Dr. Ismail:

I'm pleased to inform you that your manuscript has been deemed suitable for publication in PLOS ONE. Congratulations! Your manuscript is now with our production department. 

Kind regards, 

on behalf of

Dr. Alessandro Leone 

Academic Editor

PLOS ONE